# Exploring the Link between Work Addiction Risk and Health-Related Outcomes Using Job-Demand-Control Model

**DOI:** 10.3390/ijerph17207594

**Published:** 2020-10-19

**Authors:** Frédéric Dutheil, Morteza Charkhabi, Hortense Ravoux, Georges Brousse, Samuel Dewavrin, Thomas Cornet, Laurie Mondillon, Sihui Han, Daniela Pfabigan, Julien S Baker, Martial Mermillod, Jeannot Schmidt, Fares Moustafa, Bruno Pereira

**Affiliations:** 1Université Clermont Auvergne, CNRS, LaPSCo, Physiological and Psychosocial Stress, CHU Clermont-Ferrand, University Hospital of Clermont-Ferrand, Preventive and Occupational Medicine, Witty Fit, F-63000 Clermont-Ferrand, France; hortense.ravoux@gmail.com; 2Institute of Education, National Research University Higher School of Economics, 101000 Moscow, Russia; mcharkhabi@hse.ru; 3Psychology Department, University Hospital of Clermont-Ferrand, F-63000 Clermont-Ferrand, France; gbrousse@chu-clermontferrand.fr; 4WittyFit, F-75000 Paris, France; samuel.dewavrin@wittyfit.com (S.D.); thomas.cornet@wittyfit.com (T.C.); 5Psychology Department, Physiological and Psychosocial Stress, Université Clermont Auvergne, CNRS, LaPSCo, F-63000 Clermont-Ferrand, France; laurie.mondillon@uca.fr; 6Culture and Social Cognitive Neuroscience Laboratory, School of Psychological and Cognitive Sciences, Peking University, Beijing 100080, China; shihuih@gmail.com (S.H.); daniela.pfabigan@pku.edu.cn (D.P.); 7Centre for Health and Exercise Science Research, Hong Kong Baptist University, Kowloon Tong, Hong Kong; jsbaker@hkbu.edu.hk; 8Psychology Department, University Grenoble Alpes, CNRS, LPNC, 38000 Grenoble, France; martial.mermillod@univ-grenoble-alpes.fr; 9Emergency department, University Hospital of Clermont-Ferrand, F-63000 Clermont-Ferrand, France; jschmidt@chu-clermontferrand.fr (J.S.); fmoustafa@chu-clermontferrand.fr (F.M.); 10Biostatistics Unit, University Hospital of Clermont-Ferrand, F-63000 Clermont-Ferrand, France; bpereira@chu-clermontferrand.fr

**Keywords:** workaholism, work addiction risk, mental health, depression, quality of sleep, public health

## Abstract

*Purpose of the study*: Work addiction risk is a growing public health concern with potential deleterious health-related outcomes. Perception of work (job demands and job control) may play a major role in provoking the risk of work addiction in employees. We aimed to explore the link between work addiction risk and health-related outcomes using the framework of job-demand-control model. *Methods*: Data were collected from 187 out of 1580 (11.8%) French workers who agreed to participate in a cross-sectional study using the WittyFit software online platform. The self-administered questionnaires were the Job Content Questionnaire by Karasek, the Work Addiction Risk Test, the Hospital Anxiety and Depression scale and socio-demographics. *Data Analysis*: Statistical analyses were performed using the Stata software (version 13). *Results:* There were five times more workers with a high risk of work addiction among those with strong job demands than in those with low job demands (29.8% vs. 6.8%, *p* = 0.002). Addiction to work was not linked to job control (*p* = 0.77), nor with social support (*p* = 0.22). We demonstrated a high risk of work addiction in 2.6% of low-strain workers, in 15.0% of passive workers, in 28.9% of active workers, and in 33.3% of high-strain workers (*p* = 0.010). There were twice as many workers with a HAD-Depression score ≥11 compared with workers at low risk (41.5% vs. 17.7%, *p* = 0.009). Sleep quality was lower in workers with a high risk of work addiction compared with workers with a low risk of work addiction (44.0 ± 27.3 vs. 64.4 ± 26.8, *p* < 0.001). Workers with a high risk of work addiction exhibited greater stress at work (68.4 ± 23.2 vs. 47.5 ± 25.1) and lower well-being (69.7 ± 18.3 vs. 49.3 ± 23.0) compared with workers at low risk (*p* < 0.001). *Conclusions*: High job demands are strongly associated with the risk of work addiction. Work addiction risk is associated with greater depression and poor quality of sleep. Preventive strategies should benefit from identifying more vulnerable workers to work addiction risk.

## 1. Introduction

Workaholism is a public health concern [1,2,3] with putative deleterious health outcomes among workers [1,2,4,5,6,7,8,9,10]. Workaholism, as an unconstructive way of involvement with work, is negatively influencing employees all over the world [11,12]. Despite the important role of workaholism in the health situation of workers, this role has not been extensively studied [13]. Workaholism known as work addiction risk [1,4,14,15,16,17,18,19] is mostly defined as “a compulsion or an uncontrollable need to work incessantly” [1,2]. This internal need, known as a behavioral disorder [4], is a crucial element in identifying workaholics [3,8,9,10,16,20,21,22] and meets the general criteria of addiction [2,3,4,23].

According to Clare et al. (2014, *pp 3*) “workaholics do not engage in excessive work due to external factors such as financial problems, poor marriage, or pressure by their organization or supervisor.” Indeed, the differentiating feature a workaholic behavior from similar behaviors, such as work engagement, is the excessive involvement of the individual in work when it is not required or expected [19].

Studies show that work addiction risk is linked to positive and negative outcomes. In a negative light, work addiction risk has been found to be related to lower mental health [24,25], poorer physical health [24] and higher work–family conflict [24]. In a positive light, work addiction risk was found to be linked with an increased level of job satisfaction [7] and career satisfaction [26]. Thus, the findings show that further research is needed to shed light on this line of research. This is particularly important in France as this country is one of the industrial countries with a growing number of occupations in Europe.

Workaholic individuals are found to report more job demands in terms of work role overload and work role conflict [27]. It seems that highly demanding jobs derive individuals to become workaholic. However, the influence of work addiction risk and health-related outcomes across various occupational groups has not been much investigated. To the best of our knowledge, this association has only been examined for one occupation [8,28]. Considering the likelihood of workaholics in various occupational groups in France [14,29], this study aims to explore the link between work addiction risk and a wide range of health-related outcomes in France. Second, this study applies the framework of job-demand-control Karasek’s model (Karasek, 1979) to test the work addiction risk-health-related outcomes link across occupational groups with different degrees of job demands characterized by this model. As the job demands-control model is an occupational stress model [30], it is designed to predict negative outcomes of work stressors such as workload included in this study.

The reason for the use of this model is that this model best considers the job demands one may receive in his/her job as well as the extent to which he/she can control the job demands. Another reason is that this model has been used for job design. This means the model is assumed to be able to predict or explain the link between workaholism and health-related outcomes over occupational jobs considering their job demands in such that occupations with more demanding designs are expected to be highly linked with work addiction risk.

## 2. Theoretical Framework

In recent decades, employers have shown that they are willing to design practices and initiatives to maximize the sense of involvement of employees with their occupation with the expectation to promote their job performance [31]. Although the practices may favorably promote job performance for the employers, they also may negatively influence the mental health of employees. This is because, under such demanding circumstances, the workers may choose to spend more time at work dealing with job demands and get addicted to this working situation, known as work addiction risk. This may threaten the mental health situation of employees and results in negative outcomes [1,2,4,5,6,7,8,9,10]. This impact may even be more exacerbated by psychosocial factors. For example, Clark et al. (2016) studied the role of various psychosocial factors associated with work addiction risk and reported a negative association between age and workhalism, and a positive association between overwork climate, high job demands and reward system for high productivity with work addiction risk [32]. However, according to Taris et al. (2012) and Clark et al. (2016), a significant association between gender and work addiction risk was not found [24,31]. Moreover, they found that when the job demands are increased, the work addiction risk is more likely to be reported by workers with high job control than with low job control. Nonetheless, the associations have not been studied across various occupational groups with different job demands and job control.

We anticipate that work addiction risk might be differently experienced across various work environments; therefore, we use Karasek’s model to roll out how the design of participants’ jobs in terms of job demands and job control can lead to work addiction risk they may experience. The Karasek’s model or “Job Demand-Control-Support model” (JDCS) mainly focuses on the contradicting tasks of the work environment [33,34,35] by dividing them into the following components: job demands (i.e., all physical, psychological, social or organizational aspects of a job that require continuous physical and/or psychological—cognitive or emotional—effort [35], e.g., time constraints [36], heavy workloads [37]); job control (decision-making latitude corresponding to the use and development of one’s skills and to decision-making autonomy, i.e., ability to make its own choice for dealing efficiently with challenges [33,35,38]) and social support (from colleagues and managers) [35,39]. The JDCS model assumes four various work environments (four quadrants) in which workers may experience a different level of job demands and job control: passive, low-strain, active, and tense/job-strain [33,37,38]. “*Passive”* jobs (low job control, low job demands) might be satisfying to a worker as long as the workers reach the set goal. The passivity of a job can also be based on successfully forestalling disruptions with smooth job demands. “*Low strain”* jobs have high job control and low job demands. Individuals of this category are not particularly at risk of mental health problems, and it corresponds typically to creative jobs (i.e., architects). Workers identified as “*active*” have high job demands and high job control [33,37]. Job characteristics of those active workers are highly skilled occupations with responsibilities [33,34,37], such as heads or directors of companies. Those highly skilled workers have very demanding tasks but they have high levels of decision latitude to solve problems. Finally, workers at risk of stress-related disorders are those within the “*job strain”* group (high demand and low control). For example, health-care workers from emergency departments are typically in job strain because they cannot control the huge workload [37]. Social support modulates the impact of job strain, positively when strong social support may overcome difficulties encountered at work, such as for emergency physicians who are very solitary or negatively in the absence of social support. *Isostrain* refers to workers in a situation of job strain with no social support. According to the JDCS theory, all jobs are different in terms of job demands and job controls. Several nationwide studies on thousands of workers categorized occupations depending on the job demands and on job control [34]. It has also been suggested that workers who have a lot of authority in dealing with their strain levels participate more in working tasks. Exploring the relations between the risk of work addiction and perception of work has never been performed across various occupational categories with convenient sample size, and there are nearly no studies on the work addiction risk in the French population.

Therefore, we aimed to demonstrate the extent to which the work addiction risk is associated with the perception of work (job demands and job control), and mental health in four job categories suggested by Karasek’s model.

## 3. Materials and Methods

### 3.1. Study Methodology

This epidemiological study used a cross-sectional observational and descriptive research design. Participants were recruited through workers using the WittyFit online platform (https://wittyfit.com/) implemented in several French companies [40,41]. WittyFit is a web platform that aims to improve well-being at work. WittyFit has been designed in partnership with the University Hospital of Clermont-Ferrand in France. The National Commission for Data Protection and Liberties (CNIL), and the South-East VI ethics committee (clinicaltrials.gov identifying number NCT02596737) approved the study. Participants had no relationship with researchers. Workers through the WittyFit platform received an online questionnaire, and they had the opportunity to read the instructions and participate in the study voluntarily without further incitation from their companies or from WittyFit.

### 3.2. Participants

There were no exclusion criteria to recruit the participants. We assessed the socio-demographic characteristics of the workers. Data were collected anonymously. As this study was an online anonymous questionnaire, the need for consent was waived by the ethics committee, as long as only volunteers would answer the questionnaire.

### 3.3. Measures

**The Job Content Questionnaire (JCQ),** developed by Karasek [33,39], is an international self-assessment questionnaire which is regarded as the main instrument to evaluate three dimensions of the work environment subjectively: the job demands (nine items), the job control (nine items) and the social support (eight items) [35,37,39]. The JCQ includes 26 questions with a 4-point Likert scale from *strongly disagree (1)* to *strongly agree (4)* [34,35,37]. Higher scores indicate higher levels of psychological demands, job control and social support [35]. The thresholds are set at 21 for job demands, 70 for job control and 23 for social support [35,37]. The combination of job demands and job control can define 4 situations at work (passive, low-strain, active, tense/job-strain), corresponding to 4 quadrants [33,37,38]. In this study, we use the validated French version of the JCQ, with psychometric properties that could be verified in a French population [35]. In this study, the Cronbach’s Alphas for job demands, job control and social support were 0.58, 0.99 and 0.99, respectively.

**The Work Addiction Risk Test (WART)** was developed by Robinson et al. [1,4,41,42], from symptoms reported by clinicians caring for patients with work addiction risk [1,41]. The WART assesses 25 statements on a 4-point Likert scale from *never true (1)* to *always true (4)* [4,41,42]. The total score ranges from 25 to 100 [42] which higher scores reflecting higher work addiction risk [41]. Scores from 25 to 56 are defined as a low-risk of work addiction risk; from 57 to 66 as a medium-risk and from 67 to 100 as a high-risk [4,29,41]. We previously validated the French version of the WART [43]. In this study, the Cronbach’s Alpha was 0.90.

**The Socio-demographic characteristics questionnaire** assessed age, gender, weight, size, occupational group, education level, family situation, work characteristics, physical activity, and quantity of sleep. Sleep quality, well-being and perceived stress at work and at home are evaluated using visual analogue scales (VAS), i.e., by moving a cursor on a horizontal, non-calibrated line of 10 cm, ranging from very low (0) on the left to very high (10) on the right [44,45,46,47,48,49]. We also looked for the presence of other addictions such as tobacco, cannabis, and alcohol.

**Hospital Anxiety and Depression (HAD) Scale** is a self-report questionnaire composed of 14 items with a 4-point Likert scale from *strongly disagree (1) to strongly agree (4)* assessing anxiety (7 items) and/or depressive (7 items) symptoms [50,51,52]. For each subscale (anxiety and depression), the total score ranges from 0 to 21 [50,51]. Higher scores indicate higher levels of anxiety or depressive symptoms. A score from 0 to 7 indicates the absence of disease; a score of 8–10 represents doubtful cases, and scores higher than 11 reflect the presence of a mood disorder [50,51]. In this study, the Cronbach’s Alpha for the components of depression and anxiety were 0.82 and 0.79, respectively.

### 3.4. Statistics

The calculation of the number of subjects required for our study was carried out a priori in accordance with the literature [53] and considered the lack of any study connecting the perception of work (Karasek’s model) and work addiction risk on a heterogeneous population.

The sample size estimation was performed in order to highlight an absolute difference between “active” and “passive” workers around 20% on the prevalence of high risk of work addiction (ranging from 30% to 50% within the active quadrant of Karasek’s model), for a two-sided type I error at 5% and a statistical power greater than 90% (See Appendix A).

Statistical analyses were performed using the Stata software (version 13, StataCorp, College Station, USA). Continuous data were expressed as mean and standard deviation or median [interquartile range] according to the statistical distribution. The Shapiro–Wilk test was conducted to study the Gaussian assumption. The comparisons between independent groups (such as depending on the three groups retrieved from the WART or depending on the four quadrants from the JCQ) were carried out using ANOVA or Kruskal–Wallis test (KW) when hypotheses of ANOVA were not met (i.e., normality and homoscedasticity considered using the Bartlett test). When appropriate (omnibus *p*-value <0.05), a post-hoc test for multiple comparisons was performed: Tukey–Kramer post ANOVA and Dunn’s test after KW. The between-groups comparisons for categorical parameters were achieved with the chi-squared test or Fisher’s exact test. Relationships between variables, such as work addiction risk or job demands and job control, were studied by correlations of Pearson or Spearman depending on variables. Differences were defined as statistically significant when the level of significance (*p*-value) was less than 0.05. As proposed by some statisticians [54], we chose to report all the individual *p*-values without doing any mathematical correction for distinct tests comparing groups. Moreover, a particular focus was given to the magnitude of differences and the clinical relevance [55].

## 4. Results

### 4.1. Work Addiction Risk and Distribution of Participants

A total of 187 (11.8%) out of 1580 registered workers on the WittyFit platform agreed to respond to the WART questionnaire. Additionally, 161 workers simultaneously responded to the WART and the socio-demographic questionnaires, 183 responded to both the WART and the JCQ, and 187 to the WART and the HAD Scale (see Figure 1). Table 1 shows the socio-demographic characteristics of the workers. The 187 included workers were representative of the WittyFit users in terms of age and gender.

### 4.2. Association of Work Addiction Risk with Gender and Occupation

Of those who completed the WART questionnaire (*n* = 187), 85 (45.5%) were at low risk of work addiction, 61 (32.6%) at medium risk, and 41 (21.9%) at high risk. Women had a higher risk of work addiction than men (27% of women vs. 15% of men, *p* = 0.023). Workers with a high risk of work addiction (threshold ≥ 67 on the WART) worked an average of 7 h per week more than those at low risk (46.9 ± 13.6 h vs. 39.4 ± 10.9 h, *p* = 0.005). Workers at low, medium and high risk did not differ for other variables (Table 1).

### 4.3. Association of Work Addiction Risk and Work Perception

**Job demands and WART**: Job demands were higher for workers at high-risk of work addiction than in those at low-risk (23.8 ± 4.9 vs. 21.2 ± 7.0, *p* < 0.001). As Table 2 shows, there were five times more workers at high risk of work addiction among workers with strong job demands than in those with low job demands (29.8% vs. 6.8%, *p* = 0.002).

**Job control and WART**: There was no relationship between job control and work addiction risk. As the Table 2 shows, job control was not linked with work addiction (*p* = 0.49), and the percentage of workers with a high risk of work addiction did not change with job control (*p* = 0.77).

**Social support and WART**: There was no relationship between social support and work addiction risk. According to Table 2, social support was not linked with work addiction (*p* = 0.71), and the percentage of workers with a high risk of work addiction did not change with social support (*p* = 0.22). There was no interaction between the three dimensions of the Karasek model.

**Active workers, Job Strain and WART:** The distribution of workers with a high risk of work addiction was statistically different among the four quadrants of Karasek’s model. According to Table 3 and Figure 2, we demonstrated a high risk of work addiction in 2.6% of low-strain workers, in 15.0% of passive workers, in 28.9% of active workers, and in 33.3% of high-strain workers (*p* = 0.010). The distribution of occupations and educational levels across Karasek’s quadrants is presented in Appendix A.

### 4.4. Work Addiction Risk and Mental Health Indicators

**Depression and WART**: There were twice as many workers with a HAD-Depression score ≥11 among workers at high risk of work addiction compared with workers at low risk (41.5% vs. 17.7%, *p* = 0.009), with a significant correlation (R^2^ = 0.18, *p* = 0.014). However, as shown in Table 4, workers at low risk of work addiction were more likely to suffer from anxious symptomatology compared to workers at high risk (94.1% vs. 46.3%, *p* < 0.001), with a significant negative correlation (R^2^ = 0.60, *p* < 0.001).

**Sleep quality and WART**: Sleep quality was lower in workers with a high risk of work addiction compared with workers with a low risk of work addiction (44.0 ± 27.3 vs. 64.4 ± 2 6.8, *p* < 0.001). According to Table 4, we observed no effect on sleep quantity.

**Stress, health and WART**: Workers with a high risk of work addiction exhibited greater stress and lower health compared with workers at low risk (stress at work 68.4 ± 23.2 vs. 47.5 ± 25.1; stress at home 47.0 ± 21.5 vs. 26.3 ± 22.7; health 69.7 ± 18.3 vs. 49.3 ± 23.0; *p* < 0.001) (Table 4).

## 5. Discussion

The main findings showed that a work situation with high job demands is strongly associated with the risk of work addiction; however, the level of job control did not seem to accentuate this risk. The prevalence of work addiction risk was greater among “active” and “high strain” workers than among “low strain” or “passive” workers. Finally, work addiction had a negative association with the workers’ mental health (depression, sleep, well-being and stress).

### 5.1. Prevalence of Work Addiction

A characteristic of work addiction risk is the engagement in work in terms of hours spent [7,14,18], with workers spending the majority of their time at work and working beyond requirements [5,14,18]. Among workers who answered the WART, 22% were at high-risk of work addiction. The literature reported similar results in hospital doctors (13%) [29] or academic employees (22%) [56]. However, the prevalence of work addiction risk can be lower in other populations such as what was found in Italian teenagers (8%) [57] or with the use of other questionnaires (8%) [3]. We found a predominance of a high risk of work addiction in women (27% of women vs. 15% of men). Even if some studies did not report any gender difference in the prevalence of work addiction risk using the WART [29,56,57] or other questionnaires [3,58], most recent studies showed an increased risk of work addiction for women [59,60]. This result could be explained by the finding that women might have higher work ambitions than men [59], reflecting an evolution of women’s emancipation in our society with more involvement at work [61,62]. Perhaps this ambition of women is related to their attitude toward reaching gender equality; as such, working harder is an attempt to reduce the gap between the degree to which men and women may be differently appreciated for their services in the workplace. We demonstrated that workers with a high risk of work addiction worked 7 h more per week than those at low risk (46.9 vs. 39.4 h). Work addiction risk was not linked to other variables (BMI, physical activity, co-addiction—tobacco, alcohol, cannabis—or age and family situation), which emphasizes the major impact of the work environment on work addiction risk.

### 5.2. Work Addiction Risk and Job Demand-Control Model

We demonstrated a prevalence of work addiction risk that was five times higher in workers with strong job demands. Workers considered as active and high-strain according to the JCQ, i.e., with strong job demands in their job, were more likely to report work addiction risk. Therefore, job demands appear to be the most significant factor contributing to the development of work addiction, whereas job control and social support do not seem to have the same influence. To our knowledge, only one study has investigated the link between job perception on Karasek’s model and work addiction and found that the work addiction risk was mostly correlated with job demands. However, this study was limited to only one profession (hospital doctors) [30]. Our study confirms the major impact of workload on work addiction risk, on a larger population and on different occupational groups. Some studies have underlined the fact that work addiction risk is the consequence of an individual’s predisposing factors (personality trait, values, emotional needs) [1,4], socio-cultural experience (apprenticeship, culture of competence, organizational systems, satisfaction) [1,4,7] and family environment [1,4,7,42]. The actual work culture can result in the development of work compulsive behavior [1,9]. Working conditions are represented by the quantity and the complexity of the work, time constraints, unanticipated tasks, work disruptions, and competing demands; to put in other words by high job demands [35,39]. Thus, identifying factors such as work addiction risk is necessary to design experimental interventions or to implement preventive strategies.

### 5.3. Correlates of Work Addiction

In line with the literature, we showed that the work addiction risk was associated with mental health, in terms of depression [1,56], sleep disorders [4,5], stress [1,8,56] and well-being [1,4,8,9,17]. Workers at a high risk of work addiction had depressive but not anxious symptoms. Surprisingly, workers at low risk of work addiction had higher levels of anxiety. Indeed, some studies examined the link between anxious symptoms and work addiction and found opposite findings [41,56]. Though anxiety was assessed using another questionnaire than the gold standard (HAD scale), we have no strong explanation for the higher levels of anxiety in workers with a low risk of work addiction. Perhaps workers who report low risk of work addiction may report higher levels of work anxiety simply to be less anxious or perhaps it is because individuals have different levels of personal resources (e.g., resiliency, self-efficacy) that only allow them to get involved in the work with a certain level of job demands. Maybe workers with less personal resources are more likely to report greater anxiety. However, these assumptions would require further studies and future studies may consider them.

The positive link between work addiction risk and sleep disorders and the stress level of workers is undeniable. Sleep disorders can have a serious impact on the physical and psychological health of workers [5,63], as well as on the work itself (absenteeism, reduced productivity, and job dissatisfaction) [5]. Prolonged stress exposes workers to a high risk of systemic inflammation, metabolic disorders and coronary heart disease [37,44,46]. As studies showed, the work addiction risk can negatively be associated with family life, including neglected family, conflicts [1,4,6,14,56,64,65], which is in line with our finding demonstrating that workers with a high risk of work addiction report a high level of stress at home.

### 5.4. Theoretical and Practical Implications

From a theoretical standpoint first, this research enriches the current literature by studying the link between work addiction risk and mental health in a wide range of occupations than previous studies that studied this link only in one occupation. This more comprehensive replication provides more empirical evidence about the extent to which the work addiction risk may be detrimentally associated with mental health. Second, this study expands the literature by using the job-demand-control model as a means to explain the strength of the work addiction risk-mental health across different occupational groups. Future studies may use the same theoretical model to further explain the link between work addiction risk and mental health in other countries. Third, we used the WittyFit platform to collect data. This platform provides responders with an opportunity to see questions and respond to them using a visual analogue scale (VAS) graphically. From a practical standpoint, this study aids practitioners to identify the more vulnerable groups of workers and occupations to work addiction risk based on the job-demand-control model. Organizational decision-makers and health experts may use these findings to design interventions aiming at redesigning the jobs in such that they carry a sustainable level of job demands for the jobholders. These findings can also be used in the selection of those job candidates who will be able to deal with the job demands more efficiently. The findings revealed that workers with higher work addiction risk have greater mental health problems in terms of depression, sleep quality, and stress at work. Therefore, organizations may consider particular facilities such as access to free counseling services for the group of workers who are identified as workers with high work addiction risk.

## 6. Limitations

This study has some limitations. The response rate may seem low compared with other studies [66,67,68,69], but we included a convenience sample size of workers to demonstrate the association between the work addiction risk, job demands, and mental health-related indicators. Our sample was representative in terms of age and gender of the WittyFit users, however, companies did not give further details on the demographic profile of workers, and the selection of the WittyFit users was a convenience, non-representative sample of the French workers. Furthermore, the number of respondents in our study is in line with the literature recommendations for this type of analysis [53]. On the other hand, the homogeneous distribution of workers within the four quadrants of Karasek’s model and the strong prevalence of workers at high risk of work addiction has allowed a robust analysis. Including standard controls was not applied to our study because our study was observational and we cannot randomly assign workers in one of the four categories of the JDCS model. Finally, some professions were not represented within the professional groups we organized, which may require the expansion of our study into the general French population for future analysis. To our knowledge, previous studies on work addiction have predominantly focused on English-speaking populations [41,42,60,64]. Cultural differences could potentially play a crucial role in the development of work addiction risk [14]. Although this study is limited to a single country and its associated countries and communities, it is the first multicentric study on work addiction risk in a French population. Lastly, in this study, we used self-report scales to measure constructs such as working hours that may or may not reflect the actual time at work. Additionally, the adopted job demands scale presented a suboptimal internal consistenty. Future studies may benefit from both subjective and objective measures of this construct. Due to the cross-sectional design of this study, causal relationship between work addiction risk and mental health needs to be investigated by longitudinal studies.

## 7. Conclusions

High job demands at work are strongly associated with work addiction risk but the job control level does not play the same role. The prevalence of work addiction risk is higher for active and high-strain workers than for passive and low-strain workers. These two groups of workers appeared to be more vulnerable and therefore can suffer more from the negative outcomes of work addiction risk, in terms of depression, sleep disorder, stress and health. Preventive strategies, such as social support programs, should benefit from identifying risk factors of work addiction and vulnerable groups of workers.

## Figures and Tables

**Figure 1 ijerph-17-07594-f001:**
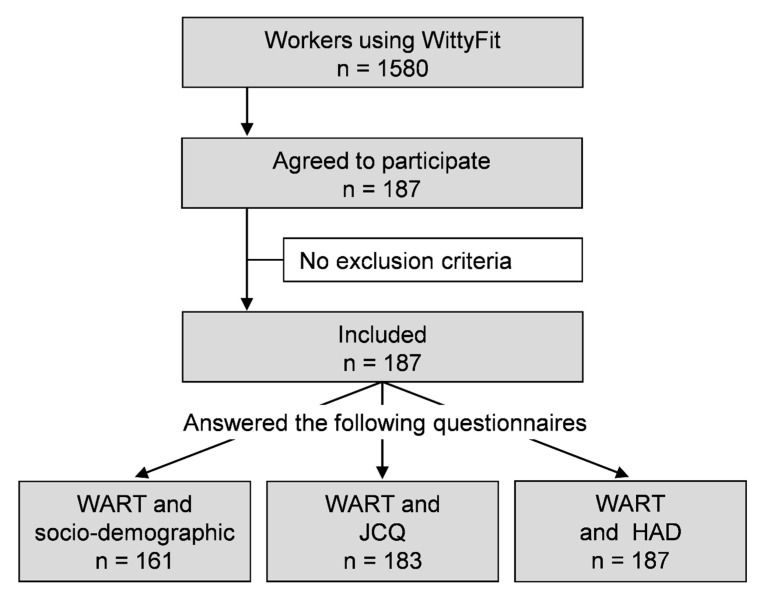
Flow chart: WART: Work Addiction Risk Test questionnaire; JCQ: Job Content Questionnaire; HAD: Hospital Anxiety and Depression scale

**Figure 2 ijerph-17-07594-f002:**
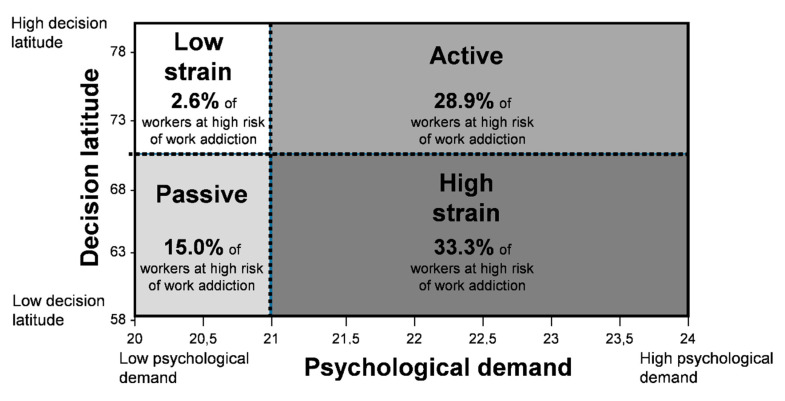
Prevalence of workers at high risk of work addiction according to Karasek’s model (Job Demand-Control model).

**Table 1 ijerph-17-07594-t001:** Workers’ characteristics based on the Work Addiction Risk Test (WART) scores.

Variables	Work Addiction Risk Test (WART)			*p*-Value
	Low-Risk Score ≤ 56	Medium-Risk 57 ≤ Score ≤ 66	High-Risk Score ≥ 67	
**Work Addiction Risk Test (WART)**				
*n* (%)	85 (45.5)	61 (32.6)	41 (21.9)	
Mean ± SD	48.1 ± 6.7	61.2 ± 2.7	73.1 ± 4.9	
**Sex**, *n* (%)				
Men	41 (56.2)	21 (28.8)	11 (15.1)	0.023
Women	31 (35.2)	33 (37.5)	24 (27.3)	
**Age (years)**, mean ± SD	43.7 ± 12.1	40.9 ± 12.0	39.4 ± 10.7	0.190
**Family situation**, *n* (%)				0.390
Single	19 (26.4)	7 (13.0)	8 (22.9)	
De facto	15 (20.8)	19 (35.2)	10 (28.6)	
Married	37 (51.4)	28 (51.9)	17 (48.6)	
Widow(ed)	1 (1.4)	0 (0.0)	0 (0.0)	
**Education level**, *n* (%)				0.210
General Certificate of Secondary Education	2 (2.8)	0 (0.0)	0 (0.0)	
A-level	3 (4.2)	4 (7.4)	1 (2.9)	
Higher National Diploma	10 (13.9)	2 (3.7)	1 (2.9)	
Bachelor degree	10 (13.9)	5 (9.3)	5 (14.3)	
Master degree or higher	47 (65.3)	43 (79.6)	28 (80.0)	
**Occupational group**, *n* (%)				0.800
Merchants–Business	2 (2.8)	2 (3.7)	2 (5.7)	
Employees	17 (23.6)	7 (13.0)	4 (11.4)	
Intermediate profession	4 (5.6)	3 (5.6)	3 (8.6)	
Inactive employment	5 (6.9)	3 (5.6)	2 (5.7)	
Manager—Intellectual profession	44 (61.1)	39 (72.2)	24 (68.6)	
**Hours worked per week**, mean ± SD	39.4 ± 10.9	40.6 ± 12.5	46.9 ± 13.6	0.005
**Seniority in the company**-years, mean ± SD	11.9 ± 10.9	11.3 ± 11.6	8.8 ± 9.31	0.330
**Body mass index**-kg.m^−2^, mean ± SD	24.4 ± 4.3	24.0 ± 4.9	24.0 ± 4.0	0.810
**Physical activity index**, mean ± SD	344 ± 355	408 ± 403	292 ± 288	0.215
**Metabolic Equivalent Task (MET)**, mean ± SD	49.3 ± 54.5	55.6 ± 55.2	40.1 ± 39.6	0.258
**Tobacco smoker**, *n* (%)	22 (56.4)	9 (23.1)	8 (20.5)	0.256
**Alcohol users**, *n* (%)	9 (30.0)	12 (40.0)	9 (30.0)	0.171
**Cannabis consumer**, *n* (%)	9 (64.3)	2 (14.3)	3 (21.4)	0.242

Notes: %, percent; *n*, number; SD, standard deviation.

**Table 2 ijerph-17-07594-t002:** Work addiction and perception of work: Scores from the Job Content Questionnaire (JCQ) and Work Addiction Risk Test (WART).

Job Content Questionnaire by Karasek	Work Addiction Risk Test (WART)			*p*-Value
	Low-Risk Score ≤ 56	Medium-Risk 57≤ Score ≤ 66	High-Risk Score ≥ 67	
**Job demands**				
Mean ± SD	21.2 ± 7.0	21.3 ± 4.6	23.8 ± 4.9	0.0007
score < 21, *n* (%)	32 (54.2)	23 (39.0)	4 (6.8)	0.002
score ≥ 21, *n* (%)	49 (39.5)	38 (30.7)	37 (29.8)	
**Job control**				
Mean ± SD	77.2 ± 12.1	77.9 ± 11.2	75.4 ± 11.0	0.499
score < 70, *n* (%)	21 (44.7)	14 (29.8)	12 (25.5)	0.772
score ≥ 70, *n* (%)	60 (44.1)	47 (34.6)	29 (21.3)	
**Social support**				
Mean ± SD	29.6 ± 41.9	31.6 ± 48.3	33.3 ± 58.9	0.709
score < 23, *n* (%)	14 (37.8)	11 (29.7)	12 (32.4)	0.220
score ≥ 23, *n* (%)	71 (47.3)	50 (33.3)	29 (19.3)	

Notes: %, percent; *n*, number; SD, standard deviation.

**Table 3 ijerph-17-07594-t003:** Prevalence of work addiction according to Karasek’s model (Job Demand-Control model).

Karasek’s Model	Work Addiction Risk Test (WART)			
	Low-Risk Score ≤ 56	Medium-Risk 57 ≤ Score ≤ 66	High-Risk Score ≥ 67	*p*-Value
Active job *n* (%)	38 (39.2)	31 (32.0)	28 (28.9)	0.010
High-strain job *n* (%)	11 (40.7)	7 (25.9)	9 (33.3)	
Low-strain job *n* (%)	22 (56.4)	16 (41.0)	1 (2.6)	
Passive job *n* (%)	10 (50.0)	7 (35.0)	3 (15.0)	

Notes: %, percent; *n*, number.

**Table 4 ijerph-17-07594-t004:** Health effects of work addiction on depression, anxiety, sleep, stress and well-being scores.

Variables	Work Addiction (WART)			*p*-Value
	Low-Risk Score ≤ 56	Medium-Risk 57≤ Score ≤ 66	High-Risk Score ≥ 67	
**HAD-Depression**				
Mean ± SD	9.2 ± 1.4	9.3 ± 1.3	10.0 ± 2.2	0.247
Score ≤ 7, *n* (%)	7 (8.2)	3 (4.9)	4 (9.8)	0.009
8 ≤ score ≤ 10, *n* (%)	63 (74.1)	49 (80.3)	20 (48.8)	
Score ≥ 11, *n* (%)	15 (17.7)	9 (14.8)	17 (41.5)	
**HAD-Anxiety**				
Mean ± SD	16.4 ± 3.1	13.9 ± 3.2	10.3 ± 3.9	<0.001
Score ≤ 7, *n* (%)	0 (0.0)	2 (3.3)	8 (19.5)	<0.001
8≤ score ≤ 10, *n* (%)	5 (5.9)	8 (13.1)	14 (34.2)	
Score ≥ 11, *n* (%)	80 (94.1)	51 (83.6)	19 (46.3)	
**Sleep** (mean ± SD)				
VAS	64.4 ± 26.8	54.4 ± 26.4	44.0 ± 27.3	<0.001
minute by night	430.4 ± 50.7	427.8 ± 63.6	415.6 ± 56.3	0.263
**VAS Stress at work**				
Mean ± SD	47.5 ± 25.1	59.4 ± 21.1	68.4 ± 23.2	<0.001
**VAS Stress at home**				
Mean ± SD	26.3 ± 22.7	39.7 ± 26.7	47.0 ± 21.5	
**VAS Well-being**				
Mean ± SD	69.7 ± 18.3	62.2 ± 21.4	49.3 ± 23.0	

Notes: %, percent; m, mean; *n*, number; SD, standard deviation; HAD: Hospital Anxiety and Depression scale; JCQ: Job Content Questionnaire; WART: Work Addiction Risk Test questionnaire.

## Data Availability

All relevant data are within the paper.

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
