# Peer review of "Exploring the Link between Work Addiction Risk and Health-Related Outcomes Using Job-Demand-Control Model"

_ijerph, 2020, doi:10.3390/ijerph17207594_

Round 1
Reviewer 1 Report
I am happy with the revised version of the paper.
Reviewer 2 Report
Thank you for the opportunity to review your manuscript "Exploring the Link between Work Addiction Risk and Health-related outcomes Using Job-Demand-Control Model".
Authors have significantly improved the quality of the manuscript.
The study is based on the JDCS theory. The literature review reads more like a list of previous research on a variety of topics rather than a theory section that explains how your different concepts are related. Try to integrate this section better and build a stronger case of the need for your study.
Authors have mainly focused on study Measures. There is less information on study methodology and participants. There is no theoretical and practical contribution to the study.
Authors have mentioned Fig 1 and Fig 2 but I can not find these two figures in the revised manuscript.
Authors should add some recent studies and cite them in the manuscript.
Please check the manuscript reference style. All the references should be according to the journal format.
The manuscript conclusion is too simple and it does not contribute to exiting studies.
